# Facelock: familiarity-based graphical authentication

Rob Jenkins[1], Jane L. McLachlan[2] and Karen Renaud[3]

[1] Department of Psychology, University of York, United Kingdom
[2] School of Psychology, University of Glasgow, United Kingdom
[3] School of Computing Science, University of Glasgow, United Kingdom

## ABSTRACT

Authentication codes such as passwords and PIN numbers are widely used to control access to resources. One major drawback of these codes is that they are difficult to remember. Account holders are often faced with a choice between forgetting a code, which can be inconvenient, or writing it down, which compromises security. In two studies, we test a new knowledge-based authentication method that does not impose memory load on the user. Psychological research on face recognition has revealed an important distinction between familiar and unfamiliar face perception: When a face is familiar to the observer, it can be identified across a wide range of images. However, when the face is unfamiliar, generalisation across images is poor. This contrast can be used as the basis for a personalised 'facelock', in which authentication succeeds or fails based on image-invariant recognition of faces that are familiar to the account holder. In Study 1, account holders authenticated easily by detecting familiar targets among other faces (97.5% success rate), even after a one-year delay (86.1% success rate). Zero-acquaintance attackers were reduced to guessing (<1% success rate). Even personal attackers who knew the account holder well were rarely able to authenticate (6.6% success rate). In Study 2, we found that shoulder-surfing attacks by strangers could be defeated by presenting different photos of the same target faces in observed and attacked grids (1.9% success rate). Our findings suggest that the contrast between familiar and unfamiliar face recognition may be useful for developers of graphical authentication systems.

# INTRODUCTION

Security codes such as passwords and personal identity numbers (PINs) are widely used to control access to resources (e.g., bank accounts, websites, mobile devices). To protect against fraudulent access, it is essential that a security code should be difficult to guess (*Garfinkel & Spafford, 1996*; *Gehringer, 2002*; *Carstens, 2009*). From this standpoint, a random sequence of symbols (e.g., "8z3gxFtv") is a much better password than a user's own surname (e.g., "jenkins"). However, security codes that are difficult to guess tend also to be difficult to remember (*Ebbinghaus, 1964*; *Craik & Lockhart, 1972*; *Zviran & Haga, 1990*; *Zviran & Haga, 1993*). For this reason, legitimate code holders are often faced with a choice between forgetting a code, which can be frustrating and inconvenient, or writing

Corresponding author
Rob Jenkins,
rob.jenkins@york.ac.uk

it down, which compromises security (*Carstens, 2009*; *Tam, Glassman & Vandenwauver, 2010*).

These and other weaknesses (*Adams & Sasse, 1999*; *Sasse, Brostoff & Weirich, 2001*) have led developers to explore other forms of knowledge-based authentication, including graphical authentication (*Blonder, 1996*; *Biddle, Chiasson & Van Oorschot, 2012*). In such systems, a user's authentication code is a set of images rather than an alphanumeric string. To log in, users identify their own images from larger challenge sets (*Podd, Bunnell & Henderson, 1996*; *Brostoff & Sasse, 2000*; *Dhamija & Perrig, 2000*; *Furnell, Papadopoulos & Dowland, 2004*; *Weinshall & Kirkpatrick, 2004*). One of the most well developed of these systems is Passfaces (*Brostoff & Sasse, 2000*), in which the images used are photographs of faces. Passfaces offers several advantages over standard passwords, most notably higher memorability of authentication codes (*Paivio & Csapo, 1973*). For example, an early evaluation found that after a 5-month delay, 72% of participants remembered their Passfaces codes on their first login attempt (*Valentine, 1998*). For comparison, a similar evaluation of passwords found that only 27% of passwords were remembered following a delay of 3 months (*Zviran & Haga, 1993*). In a pioneering field trial, *Brostoff & Sasse (2000)* reported that login failures were three times higher for passwords than for Passfaces. This estimate is consistent with previous findings. However, such graphical systems are not without their limitations (*Furnell, Papadopoulos & Dowland, 2004*; *Tari, Ozok & Holden, 2006*; *Everitt et al., 2009*; *Mihajlov & Jerman-Blazic, 2011*). Perhaps foremost among these is their susceptibility to 'shoulder-surfing' attacks (*Tari, Ozok & Holden, 2006*), in which an attacker obtains a user's authentication code by secretly watching the user during authentication. This attack is powerful because it exploits the memorability of image-based codes: images that are easy for the user to recognise are also easy for an attacker to recognise (*Paivio & Csapo, 1973*).

In the present study we show that this symmetry—between ease of recognition for the user, and ease of recognition for the attacker—can be broken by applying insights from cognitive psychology research. Psychological studies of face recognition have revealed strong qualitative differences between processing of familiar and unfamiliar faces (*Burton & Jenkins, 2011*; *Jenkins & Burton, 2011*). When a face is familiar to the viewer, it can be identified from a wide range of different photographs, even when image quality is very poor (*Harmon, 1973*; *Burton et al., 1999*; *Burton, Jenkins & Schweinberger, 2011*; see Fig. 1). Importantly for this study, different images of a familiar face are almost never mistaken for different people (*Jenkins et al., 2011*). In contrast, our ability to identify unfamiliar faces from photographs is strikingly poor (*Bruce et al., 1999*; *Bruce et al., 2001*). Very often, different photos of an unfamiliar face are seen as different individuals (*Jenkins et al., 2011*). Thus, familiarity with a particular face determines one's ability to identify it across changes in image (see Fig. 2). Although the transformative effect of familiarity on face recognition may be not be intuitively obvious, it is highly robust, and has been replicated in dozens of experiments spanning decades of research (*Bruce, 1982*; *Clutterbuck & Johnston, 2002*; *Clutterbuck & Johnston, 2004*; *Clutterbuck & Johnston, 2005*; *Megreya & Burton, 2006*; *Jenkins et al., 2011*).

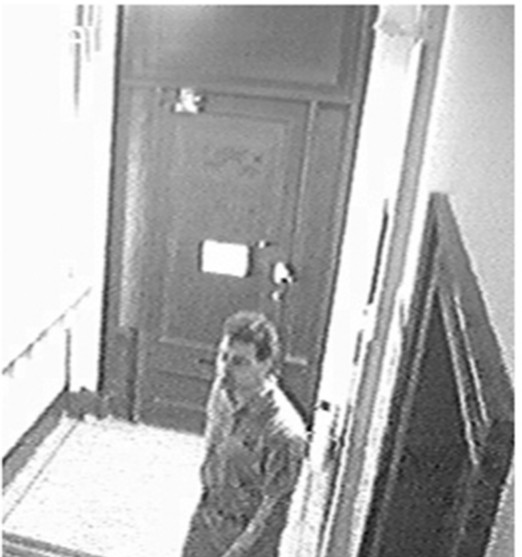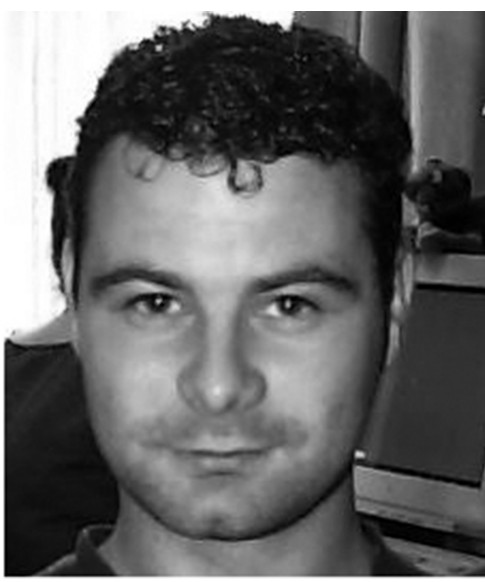

**Figure 1** **Matching a face to a poor quality CCTV image.** Example images from *Burton et al. (1999)*. Matching poor quality images is easy for observers who are familiar with the faces concerned. Performance of unfamiliar observers is strikingly poor. These images both show the same person.

The familiarity contrast is normally encountered as a *problem* in applied settings. For example, unfamiliar face matching presents a serious challenge for security personnel and for automatic face recognition systems. In the present study we offer a very different perspective by describing how the familiarity contrast might be exploited positively as the basis of an authentication system. The principle is straightforward: familiarity with a particular face determines an observer's ability to identify it across different photographs. For any individual face that is not widely known, this ability will be very narrowly concentrated within the population. If a *set* of such faces is known only to a single individual, it can be used to create a personalized lock. Access is granted to anyone who demonstrates image-invariant recognition of the critical faces, that is, anyone who is familiar with them all. Conversely, access is denied to anyone who does not demonstrate image-invariant recognition of the critical faces, that is, anyone who is not familiar with them all.

To test this principle, we developed a prototype system that involves presenting a series of face arrays, similar to Passfaces. In our scheme, each array contains one face that is familiar to the user, among other faces that are unfamiliar. The user gains access by simply indicating the familiar face in each array. We refer to this method as Facelock. The scheme has two major advantages over traditional authentication methods. First, there is no explicit memory involved—the task is simply to pick out the familiar face in each array. As this task does not require the user to remember a code, the issue of forgetting one's code does not arise. Dispensing with a set code also means that the challenge arrays, and the familiar faces embedded in them, may be composed of different photographs of different individuals at each login. This is very different from the traditional approach of assigning a single invariant authentication code to an account holder. The second major advantage

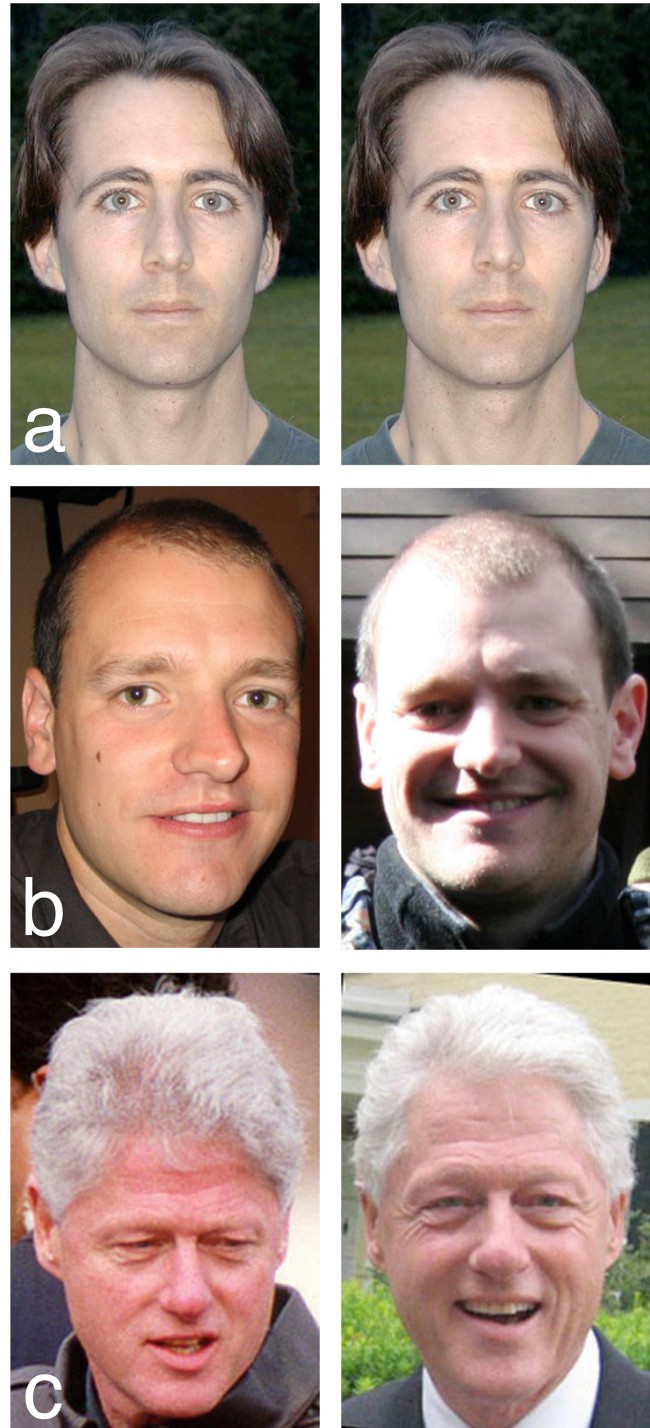

**Figure 2 Familiar and unfamiliar face matching.** (A) Matching identical images is trivial. (B) Matching different images of unfamiliar faces is hard. (C) Matching different images of familiar faces is easy.

concerns the problem of writing down authentication codes (*Dunphy, Nicholson & Oliver, 2008*; *Tam, Glassman & Vandenwauver, 2010*). In the proposed scheme, the user is not required to reproduce a particular set code in order to authenticate. The only requirement is to distinguish familiar faces from unfamiliar faces. As familiarity discriminations are extremely robust (*Young, Hay & Ellis, 1985*; *De Haan, Young & Newcombe, 1991*), users have no incentive to write down *aide-memoires* for their targets, and the associated security risk can be avoided.

The main aims of the current work are (i) to test the feasibility of an authentication method that exploits the familiarity contrast in face recognition, and (ii) to assess its resilience against two very different forms of attack—guessing by high-acquaintance attackers, and shoulder-surfing by zero-acquaintance attackers. The aim at this stage is not to develop a commercially viable system. Instead we seek to raise awareness of the important psychological contrast between familiar and unfamiliar face processing, and to explore the potential for exploiting this contrast in the context of authentication. We begin in Study 1 by comparing authentication rates for legitimate account holders with authentication rates for (i) zero-acquaintance attackers, and (ii) personal attackers who know the participants very well (e.g., spouses, family members). In Study 2 we examine whether a full-visibility shoulder-surfing attack can be thwarted by presenting different photographs of the same targets to the participant and the attacker.

## STUDY 1

The main aim of the first study was to establish whether participants could in practice generate suitable target faces. These should be faces that the participants know well, so that they could easily recognise them from photographs, but that other people do not know well, so that *all* of the faces in the array are unfamiliar to potential attackers. If such targets can be found, then it should be possible to differentiate between account holders and attackers by comparing target detection performance. To anticipate, we found that suitable targets were readily volunteered by participants. Authentication rates were very high for legitimate users, even after a delay of one year. In contrast, authentication rates for attackers were very low, even when the attackers were close acquaintances of the users.

## METHOD

### Participants

A total of 396 volunteers contributed data. 120 were volunteers who responded to our recruitment email (54 male, 66 female; age range 18–79). These 120 volunteers served as account holders in the current study. A further 110 volunteers were recruited from our participant pool to act as zero-acquaintance attackers, that is, people who knew nothing about the account holders. For comparison, we also asked each account holder to nominate two close acquaintances (e.g., spouses, family members) who could act as personal attackers. We reasoned that the faces of people who are familiar to participants might also be familiar to their close acquaintances, giving these personal attackers a significant advantage. We acknowledge that this personal attacker selection is unrealistic, as

**Peer**J

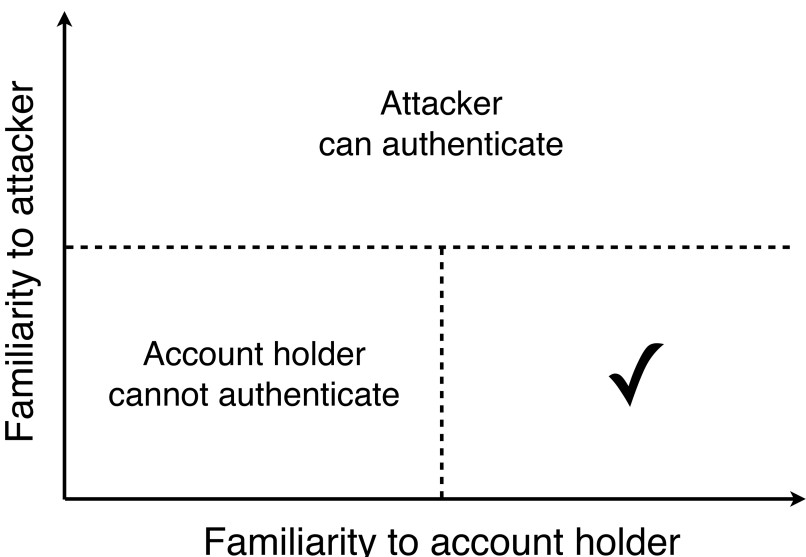

**Figure 3** **Selecting Facelock targets.** A schematic diagram summarising the requirements of target faces. If the target is familiar to the attacker, the attacker will be able to authenticate. If the target is unfamiliar to the account holder, the account holder will be unable to authenticate. The tick represents the region of acceptable targets.

it assumes that attacks only ever come from close acquaintances, and never from strangers. However, we prefer here to underestimate the security of the system than to overestimate it. 166 nominated attackers took part. All account holders and nominated attackers were offered entry into a prize draw for an iPod Nano. The study received ethical approval from the FIMS Faculty Ethics Committee at the University of Glasgow (CSE 00871).

## Design and procedure

The study consisted of seven distinct phases—three preparation phases and four test phases. We describe each of these below.

### *Phase 1: target nomination*

Account holders nominated four or more target faces by entering the targets' names on the project website. As the proposed system relies on account holders and attackers having contrasting degrees of familiarity with the targets, appropriate selection of targets was critical. Ideally, an account holder's targets should be well known to the account holder, but unknown to other people. Our pilot work indicated that it can be difficult spontaneously to generate targets that satisfy both of these requirements. For this reason, we provided account holders with the following instructions in order to guide them to the appropriate region of their search space. Figure 3 represents the constraints on target selection schematically.

"The next page will ask you to list some minor celebrities—*really* minor celebrities. Almost everyone recognises the 'A-List' celebrities below [photos of international celebrities such as major film stars]. Most people also recognise some 'B-List' celebrities

[photos of national celebrities such as television presenters]. We want you to tell us your 'Z-List' celebrities.

By 'Z-List' celebrities, we mean people who are (or were):

1. Only famous within a narrow field of interest. For example, a famous skier or a famous cellist. This could include someone who was famous many years ago, but who is not well known these days.

2. Well known to you, so that you would easily recognise them from photographs.

3. Not well known to the public at large, so that you would not expect others to recognise them.

4. Possible to find using a Google Image search.

This is the most challenging part of the study, but also the most important."

Having read these instructions, account holders were asked to submit the names of four or more targets, up to a maximum of ten. There was no time limit for this task, and account holders were free to log out and return later to complete it. Once an account holder was satisfied with this personal list, the names were transferred to the experimenter. Each account holder was also asked to provide email addresses of two close acquaintances (e.g., spouses, family members) who would be willing to act as personal attackers.

### Phase 2: image collection

Targets who had already been nominated by another account holder (<1%) were eliminated to avoid ambiguity at login. For all other targets, the experimenter collected at least four face photographs by using the target's name as a Google Image search term. We accepted the first four photographs in which the whole face was visible, regardless of viewing angle, lighting, age, or other sources of image variability. This resulted in 4 different photographs for each of 603 faces (2412 images in total). All photos were cropped to a rectangular frame measuring 100 pixels wide ×119 pixels high for presentation. The collected photos of each account holder's targets were then uploaded to the project website for that account holder to approve.

### Phase 3: image approval

Account holders returned to the website to view the photos of their targets and to approve or decline each image. The purpose of this step was twofold. First, it allowed us to ensure that the photos depicted the correct individual. This was necessary as names are rarely unique identifiers, and search results invariably included images of more than one person. Second, it allowed us to confirm that the returned images were indeed recognisable to their nominators. Declined images (<1%) were replaced until the account holder was satisfied with the selection. Image approval was followed by a delay of one week to allow forgetting of the selection procedure. Account holders then received an email requesting them to return to identify their faces again.

### Phase 4: account holder login (one week delay)

After the one-week delay, account holders returned to the project website and attempted to authenticate. The account holder's lock consisted of a series of four different challenge

sets, each comprising nine face photographs arranged in a 3 × 3 grid (similar to Passfaces challenge sets; see Fig. 4). In each grid, one image (the target) was a random photo of a person selected at random from that account holder's pool of target names. The remaining eight images (the distractors) were random photos of faces drawn at random from other account holders' pools of targets. Allocation of the nine images to the nine grid positions was randomised so that location was not predictive of target/distractor status. This meant that from the perspective of the account holder, each grid contained one familiar face among eight unfamiliar faces. However, from the perspective of an attacker, all nine faces should be unfamiliar. The account holder's task was simply to click on the familiar face in each grid. Identifying the correct image in all four grids resulted in successful authentication. The probability of opening the lock by chance alone was thus 1 in 6561, or 0.015%, for this particular instantiation.

No feedback was given until the end of the four-grid lock, after which the account holder was told whether or not the authentication attempt was successful. If the attempt was unsuccessful, the lock was reset using newly selected photos, and the account holder was asked to try again. Following successful authentication, or three unsuccessful attempts, the account holder proceeded to a brief questionnaire concerning account holders' impressions of the system.

### Phase 5: zero-acquaintance attacker entry

In small-scale pilot studies, we found that medium-acquaintance attackers (work colleagues) were never successful. To estimate the success rate in a larger sample, we recruited 114 zero-acquaintance volunteers to attack a randomly allocated lock. These 114 volunteers undertook 207 attacks between them. The authentication procedure for the attacker phase was exactly the same as for the account holder phase, with one of the account holder's targets and eight non-target faces making up each grid. As with the account holder entry phase, no performance feedback was given until successful authentication, or three unsuccessful attempts. We expected that if the account holder chose appropriate targets, none of these faces should be familiar to the attacker, and the success rate should not exceed chance levels. The zero-acquaintance attackers were recruited to verify that this was the case. However, our main interest was in the success rate of the personal attackers.

### Phase 6: personal attacker entry

In the first phase of the study, each account holder was asked to provide email addresses of two close acquaintances who would be willing to act as personal attackers. A total of 166 personal attackers agreed to take part, undertaking 249 attacks between them. Importantly, attackers only attacked their own nominator, so that every attack was from a close personal acquaintance of the account holder (e.g., spouse, family member), rather than from a stranger. Again, the authentication procedure was the same as for the account holder phase. If the account holder chose appropriate targets, all of the faces in all of the grids should be unknown to the attacker. We reasoned that high-acquaintance attackers might have acquired a degree of familiarity with their nominators' targets, due to shared

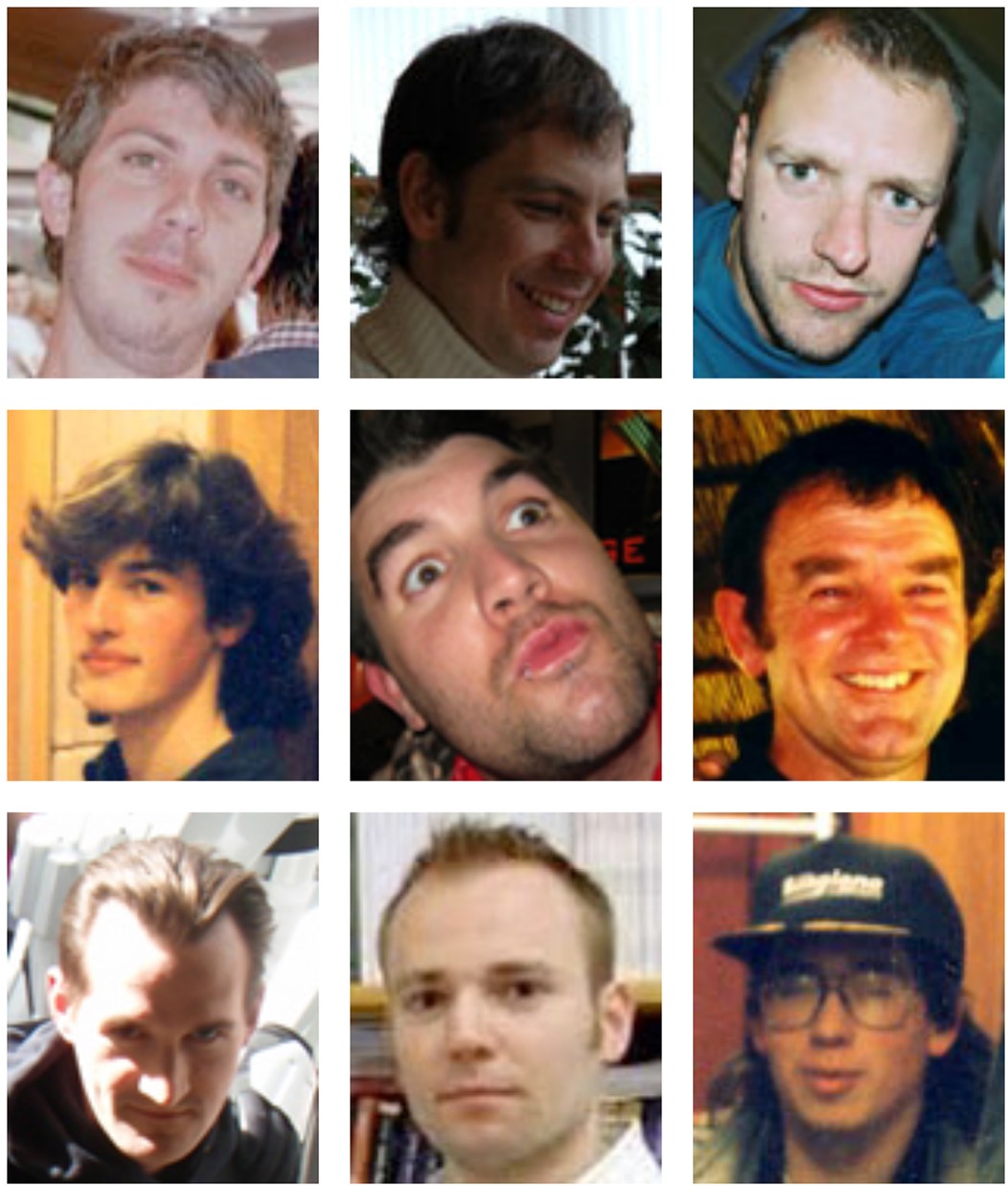

**Figure 4 A challenge grid in Facelock.** An example grid consisting of one face (the target) that is familiar to one of our account holders, and eight faces that are unfamiliar to the same account holder. Readers are invited to guess which of the nine faces is the target. For someone who doesn't know the account holder, it is difficult to find any basis for this decision.

exposure (e.g., overlapping interests or media consumption), thus providing a more stringent test. As with the account holder entry phase, no performance feedback was given until the end of the entire four-grid sequence that comprised a single lock. Following successful authentication, or three unsuccessful attempts, the attacker proceeded to a brief questionnaire concerning attackers' impressions of the system.

**Table 1** **Authentication rates in Study 1.** Shown separately for account holders and attackers. See main text for details of delays of procedure.

| | N | Succeeded | | Succeeded (1st attempt) | | Failed | |
|---|---|---|---|---|---|---|---|
| Account holders (1 week delay) | 120 | 117 | 97.5% | 101 | 84.1% | 3 | 2.5% |
| Account holders (1 year delay) | 79 | 68 | 86.1% | 62 | 78.5% | 11 | 13.9% |
| Zero-acquaintance attackers | 114 | 1 | 0.9% | 0 | 0% | 113 | 99.1% |
| Personal attackers | 166 | 11 | 6.6% | 5 | 3.0% | 155 | 93.4% |

***Phase 7: account holder login (one year delay)***

One year after the initial account holder login phase, account holders were asked to authenticate a second time. This was the only contact between experimenters and account holders since the initial login phase, and our log confirmed that none of the participants had visited the project website during the intervening months. Thus, the one year interval provided an excellent opportunity for account holders to forget about the study (*Ebbinghaus, 1964*). Previous research has shown that passwords are quickly forgotten once they fall into disuse (*Witty & Brittain, 2004*). For example, two studies of password memorability (*Zviran & Haga, 1990*; *Zviran & Haga, 1993*) reported memorability rates of 35% and 27.2% after a delay of five months. Given that a putative advantage of our familiarity-based approach is that it imposes no memory load, we predicted relatively preserved authentication rates even after a year of disuse.

## RESULTS AND DISCUSSION

### Authentication data

As can be seen in Table 1, 97.5% of account holders (117/120) successfully authenticated, with 84.2% (101/120) succeeding on the first attempt. In contrast, only 6.6% of personal attackers (11/166) were successful, and only 3.0% (5/166) on the first attempt. This compares favourably with previous analyses based on Passfaces (*Davis, Monrose & Reiter, 2004*).

Chi Square analysis of these total success rates confirmed a highly significant difference between account holders and personal attackers [$\chi^2(1) = 232.6$, $p < .0001$]. We also note that the majority of account holders' failures to authenticate were 'near misses', in which three of the four targets were correctly identified. For personal attackers, near misses were the least frequent authentication failure.

Only one attack by a zero-acquaintance attacker was successful, precluding any statistical analysis for this group. However, the circumstances of the one successful attack are perhaps revealing. Specifically the account holder had not chosen 'Z-List' celebrities as required. Indeed, for the successfully attacked lock, two of the four faces were members of the rock band Led Zeppelin (Robert Plant, Jimmy Page), perhaps analogous to choosing "ledzeppelin" as a password.

Analysis of the 11 successful attacks by nominated attackers revealed similar regularities. In five of these cases, the account holders had chosen widely-known celebrities as targets (e.g., Tony Blair, John Wayne), instead of 'Z-List' celebrities. In a further three cases,

the account holders were non-Caucasian, and chose only non-caucasian target faces. Since virtually all of the distractor faces were Caucasian, these account holders' targets were presumably easy for their nominated attackers to guess. Nominated attackers were always close acquaintances of the account holders in this study, and so knew the ethnic background of the account holders they were attacking. For the remaining three successful attacks, we suggest that the attackers had some degree of familiarity with their account holders nominated targets—enough to set the targets apart from the distractors. For example, musicians that one likes might be recognized by one's spouse, due to shared exposure.

Taken together, the success rates of account holders (97.5%), randomly zero-acquaintance attackers (<1%), and nominated high-acquaintance attackers (6.6%) strike us as a promising starting point. Analysis of successful attacks provides little evidence that the principle of exploiting familiarity contrast is problematic. Rather, the main challenge is the separable problem of compliance: if the system is not used as intended, it does not work as well. This limitation is characteristic of a wide range of security systems—including passwords, PIN codes, and mechanical locks.

## Delayed authentication

79 of our initial account holders returned to login a second time, following a one-year delay. As can be seen in Table 1, 86.1% of these returning account holders (68/79) successfully authenticated, 78.5% (62/79) on their first attempt. This is a remarkably well-preserved success rate over such a long period of disuse, especially given that different images of the account holders' targets were presented at the delayed login. For comparison, previous research reported a first-attempt authentication rate of 77% after only two weeks when using traditional passwords (*Bunnell et al., 1997*). Established graphical authentication systems are also vulnerable to memory decay, though generally to a lesser degree than passwords. One influential study (*Valentine, 1998*) reported an authentication rate of 72% (by third attempt) after a five-month delay when using Passfaces. Although these comparisons involve rather different authentication methods, they highlight the very different demands of recall-based, recognition-based, and familiarity-based decisions.

We attribute account holders' high success rate in the present study to two main factors. First, there was no authentication code to remember, so the classic problem of account holders forgetting authentication codes did not apply. Second, our account holders had already established robust mental representations of their target faces prior to the study (they were familiar faces), so presenting different images of these targets did little to impede recognition (*Jenkins & Burton, 2011*). Interestingly, a number of returning account holders commented on the surprising ease of authentication under these conditions. One wrote, "I didn't think I could log in because I couldn't remember any of the people I chose—but I did!" Interestingly, another reported, "I got them all right. Did you use the same images of the people or different ones? I got the impression that I did not recognise the image but the person".

**Table 2 Questionnaire data from Study 1.** Percentage 'Yes' responses for account holders' questionnaire items from Study 1.

| | |
|---|---|
| I wrote my targets' names down to remember them. | 0% |
| I found it hard to identify my target faces. | 10% |
| Upon reflection, I would have chosen different target faces. | 80% |
| I was confused by recognising more than one face in a grid. | 16% |
| I would be prepared to use a system like this to log in rather than a password. | 31% |

**Table 3 Attacker questionnaire data from Study 1.** Mean Likert scale ratings (1–5) for personal attackers' questionnaire items from Study 1.

| | |
|---|---|
| How much effort was involved in guessing the targets? | 2.9 |
| How hard was it to put yourself into the account holder's shoes to guess his/her targets? | 3.5 |
| How successful do you think you were? | 2.3 |
| How well do you know the person? | 4.4 |

## Account holders' questionnaire data

Account holders responded to five questionnaire items concerning user experience. Summaries of these responses can be seen in Table 2.

The questionnaire data contain little evidence that account holders had difficulty using this system. None of the account holders reported writing down their targets' names. This suggests that they correctly understood that forgetting their targets was not an issue. Only 10% of account holders reported difficulty in identifying their target faces. Thus most account holders were successful in nominating faces that they could recognise well. Interestingly, the great majority of account holders (80%) stated that with the benefit of hindsight, they would have chosen different targets. Presumably, since account holders had little trouble recognising targets that they actually chose, their motive here was not making authentication easier for themselves, but making it harder for attackers. 16% of account holders reported recognising one of the non-target faces in a grid. However, the overall authentication rate of 97.5% implies that this confusion rarely stopped them from authenticating correctly. On the basis of this experimental trial, 31% of participants said that they would use a Facelock system instead of a password, 25% said they would not, and 44% were undecided. Given that we made no concessions to usability and HCI issues in this study, it is perhaps surprising that 31% of respondents were positively disposed to the method.

## Personal attackers' questionnaire data

Personal attackers responded to four questionnaire items using a 5-point Likert scale, where 1 indicates a low rating, and 5 indicates a high rating. Mean ratings for each item are shown in Table 3.

Personal attackers found guessing their account holders' targets moderately effortful, and found it quite difficult to imagine who the account holder might have chosen.

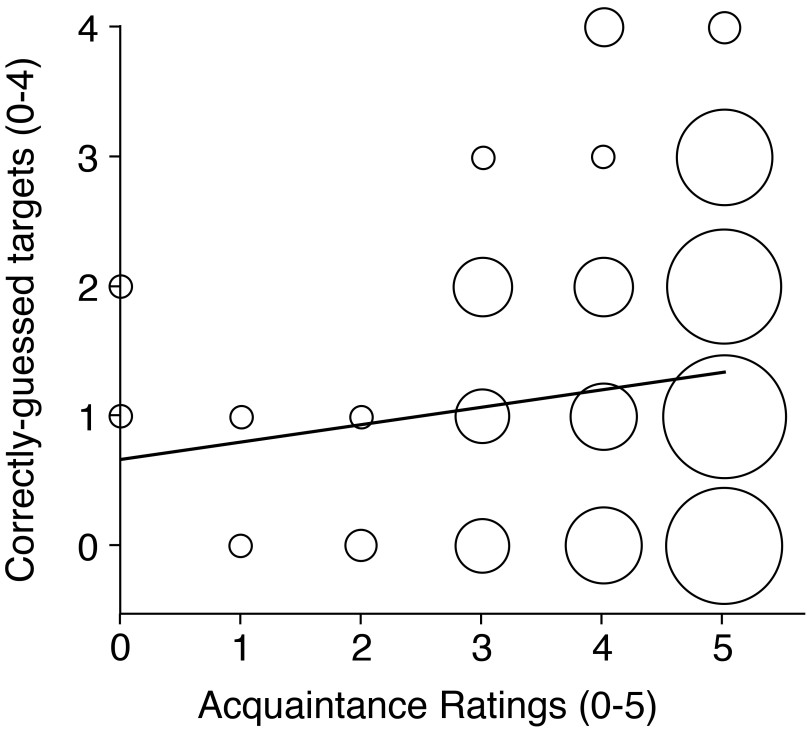

**Figure 5** **Attack success as a function of personal acquaintance.** Scatterplot showing the relationship between personal attackers' acquaintance ratings and the number of correctly guessed targets in their first attacks. The area of each datapoint is proportional to the number of cases contributing to it.

Consistent with these impressions, they rated their level of success as rather poor overall, though even this rating is a generous appraisal of their actual success rate. Personal attackers knew their account holders very well overall, confirming good compliance among account holders at the attacker nomination stage. To test whether personal attackers were more successful the better they knew their victims, we computed the correlation between these attackers' acquaintance ratings for Item 4 above, and the number of correctly-guessed targets (0–4) in their first attacks (see Fig. 5).

This correlation was moderately positive and highly reliable [$R = 0.29$, $N = 166$, $p < 0.001$]. Importantly, lower acquaintance attackers (ratings <4) were never successful. We return to the issue of acquaintance in the General Discussion section.

## STUDY 2

The preceding study confirmed that account holders who were familiar with the target faces could easily distinguish these faces from unfamiliar non-targets, regardless of the particular photos that were used to portray them. In contrast, attackers found it very difficult to guess account holders' targets, even when the attackers were close acquaintances of the account holders.

The second study focuses on a different aspect of the proposal, specifically the use of multiple photos of each target. We also sought to compare the resilience of different

account holders' locks directly, by exposing them to multiple attacks. To this end, we modelled a best-case scenario for shoulder-surfing attacks, in which we presented the correct authentication sequence to attackers under full-visibility viewing conditions, and then asked them immediately to replicate the sequence using different photographs of the same target faces. Attackers were thus required to generate the sequence of identities that they had just seen, even though those identities were portrayed using different images.

As in Study 1, we loaded this situation heavily in the attackers' favour. First, we used the same four target identities for the observation sequence and the replication sequence, rather than drawing a set of four targets at random from the account holders' entire pool. Second, we presented these same four targets in the same order in both sequences, rather than presenting them in a different random order each time. Third, attackers did not have to glance furtively at the authentication sequences for fear of being noticed. Instead, we presented the sequences very clearly to the attackers, who were asked to give it their full attention. Finally, there was no delay between the observation sequence and the replication task. Thus attackers' memory decay was minimized. These real world complications were eliminated in an effort to isolate the impact of a photo change. It is already established that replicating a four-item sequence is well within the limits of human short-term memory. This is true in experimental settings (*Miller, 1956*), and also in the context of shoulder-surfing 4-digit PIN numbers (*Anderson, 1993*). However, the present case differs from previous studies in that different images of each item are used at the sequence replication stage. If attackers are able to integrate across different photos of each target efficiently, then performance should be close to ceiling (*Miller, 1956*). Alternatively, if a change in photograph impedes identification in this situation, then performance should be relatively poor, even when the authentication code is clearly presented to the attacker immediately before the attack.

## METHOD

### Participants

Thirty-two postgraduate volunteers (6 male, 26 female; age range 21–36) completed the study. The study received ethical approval from the FIMS Ethics Committee at the University of Glasgow.

### Design and procedure

Each participant attacked five locks so that each lock was attacked 32 times. The five locks (i.e., 5 different 4-grid sequences) were drawn at random from those that led to successful authentication by account holders in Study 1. In other words, the authentication rate for account holders was 100% for this sample of locks. For each lock, a different-image version was also constructed, by replacing the target from each grid with a different photo of the same person, and replacing the eight non-targets with different non-targets.

As with the original grids, the location of the images in the grid was randomised. To make the task as easy as possible for the attackers, grid order was preserved across observation and replication sequences, so that the same targets appeared in the same order

Table 4 **Shoulder-surfing data from Study 2.** Columns refer to the different locks, and rows refer to the number of correctly-guessed targets. All four targets must be correctly guessed for the attacker to gain entry.

| Correctly-guessed targets | Lock 1 | Lock 2 | Lock 3 | Lock 4 | Lock 5 |
|---|---|---|---|---|---|
| 0 | 9 | 0 | 5 | 15 | 9 |
| 1 | 10 | 4 | 14 | 10 | 13 |
| 2 | 11 | 16 | 12 | 7 | 9 |
| 3 | 2 | 9 | 1 | 0 | 1 |
| 4 | 0 | 3 | 0 | 0 | 0 |

(1–4) in both versions of the lock. The different-image versions of the grids were printed at a size of 10 cm × 12 cm and bound into response booklets. The original grids were projected at a size of 150 cm × 180 cm using a computer controlled data projector, which attackers viewed at a distance of between 3 and 5 metres.

For each of the five locks, attackers first watched the authentication sequence using the original grids, and then tried to replicate the sequence on the different-image grids, that is, to copy the account holder's authentication code. To demonstrate each sequence as clearly as possible, each one of the four grids was presented on screen for 5 s together with its grid number (1–4). After the first 2 s, a green frame appeared around one of the faces, identifying that face as the target (analogous to watching the account holder select that face). As face identification is normally accomplished within about 200 ms of stimulus onset (*Liu, Harris & Kanwisher, 2002*), we expected this presentation time to allow full encoding of the correct target. This procedure was intended to model observation of target selection in an optimal shoulder surfing situation, in which all the necessary information is presented clearly at the focus of attention. Readers are invited to simulate this task for a single grid by comparing Figs. 6 and 4.

Successive grids in each lock were separated by a blank interval of 2 s. Immediately after the fourth target had been revealed, attackers were asked to reproduce the sequence they had just seen, by circling the same four targets on their response sheets. There was no time limit for this task. When the attackers were ready to proceed (<60 s in all cases), the next authentication sequence was initiated. All 32 participants attacked the same 5 locks once, resulting in 160 attacks in total.

## RESULTS AND DISCUSSION

Raw frequency data are shown in Table 4. Only 3 out of 160 attacks were successful (1.9%). This strikes us as a very promising figure, especially given the privileged conditions of attack. When attempting to replicate the authentication sequence, attackers saw the same targets presented in the same order under highly favourable viewing conditions and with no time pressure. Only the photo used for each face was changed. As it turned out, this alone was enough to defeat these shoulder-surfing attacks.

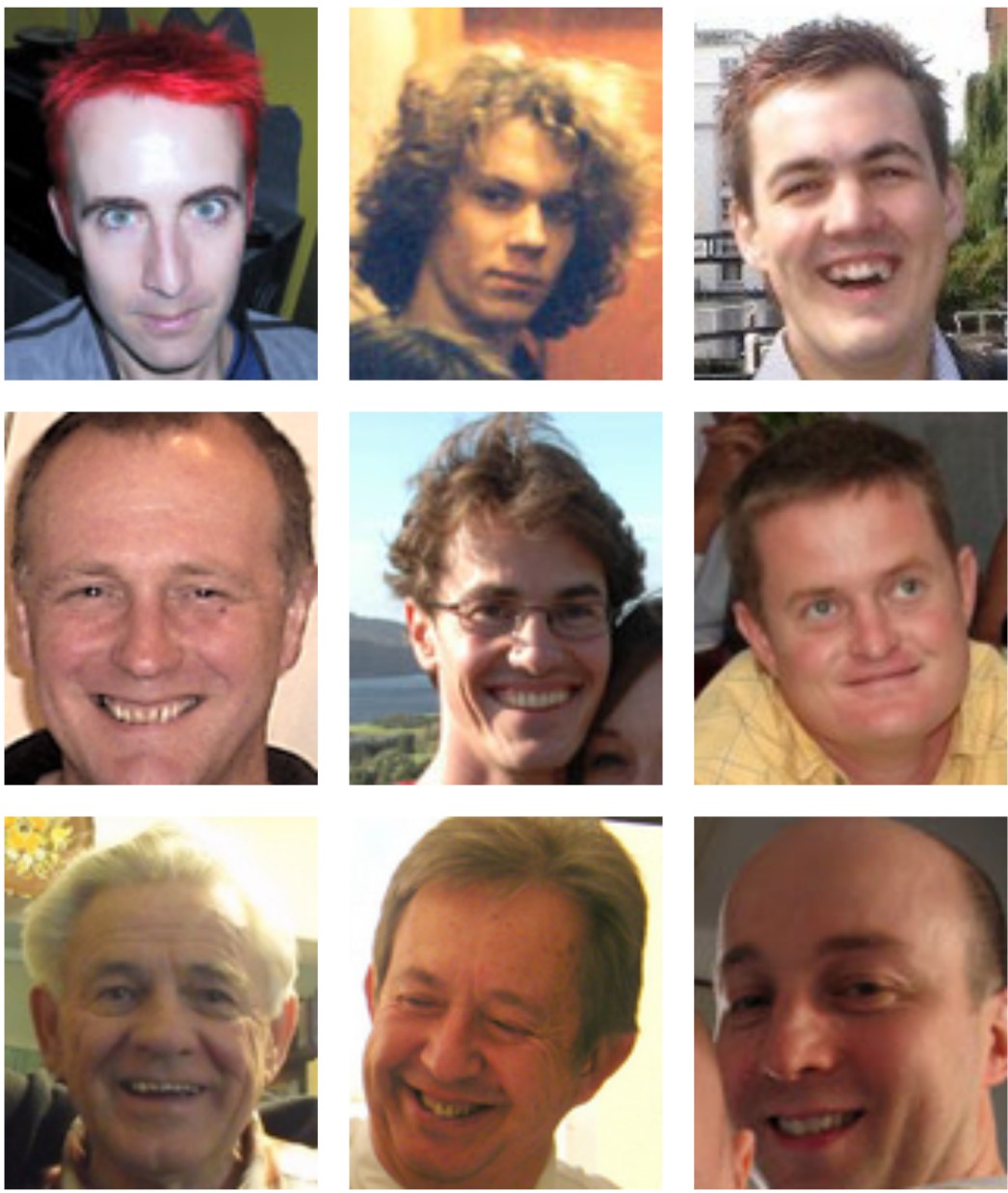

**Figure 6  A second challenge grid.** One of these faces is also present in Fig. 4. Even with a single grid, it is difficult to determine which face is repeated simply by trying to memorise Fig. 4. Side-by-side matching of unfamiliar faces is also highly error prone (*Jenkins & Burton, 2008*; *Jenkins & Burton, 2011*).

We note that all three successful attacks were on the same lock. Inspection of the targets in this particular lock suggests that this may be due to their distinctive appearance. For example, one of the targets was bald and wore glasses in both photos; another was an elderly woman with permed white hair. As none of the distractor faces shared these features, the matching targets were presumably rather salient in this context. In the General Discussion we consider how this situation could be avoided.

## GENERAL DISCUSSION

### Summary of findings

Two studies tested a knowledge-based authentication system that exploits the psychological contrast between familiar and unfamiliar face recognition. In Study 1 we found that account holders were able to generate target faces that were well known to themselves, but were not well known to other people. Account holders authenticated easily by detecting these familiar targets among other faces (97.5% success rate), and this was the case even after a one-year delay (86.1% success rate). By contrast, zero-acquaintance attackers were reduced to guessing (<1% attacks rate). Even personal attackers who knew the account holder well were rarely able to authenticate (6.6% success rate). This success rate for attacks compares favourably with previous studies. Analysing a system based on Passfaces, *Davis, Monrose & Reiter (2004)* conclude that 10% of authentication codes could be guessed within one or two attempts, even by very low acquaintance attackers who know only the gender or race of the account holder. Here we found a successful attack rate of 6.6% within three attempts for very high acquaintance attackers who knew a great deal about the account holder. In Study 2 we found that optimal shoulder-surfing attacks by strangers could be repelled simply by using different photos of the targets in the observed and attacked grids (1.9% success rate). Together, these findings suggest that the contrast between familiar and unfamiliar face recognition may be useful for graphical authentication systems. Although face-based systems have been developed previously, these have always conflated face recognition and image recognition, by representing each face with a single image (*Jenkins et al., 2011*). As image memory will be equally excellent for account holders and attackers, such systems are vulnerable to shoulder-surfing attacks (*Tari, Ozok & Holden, 2006*). The use of different photographs for each target confounds attackers who are unfamiliar with the targets, but does not impede legitimate users who are familiar with their chosen targets.

The approach we describe here offers two advantages. First, unlike a conventional password, it does not require the account holder to remember anything specific to the authentication procedure, as the task is simply to indicate which of several faces is familiar. The system thus exerts very little memory load compared with conventional passwords. Our most striking evidence for this comes from the delayed authentication task in Study 1. Here, account holder's authentication rate was 86%, one year after a single login. This is unprecedented for knowledge based authentication systems (*Sasse, Brostoff & Weirich, 2001*). For comparison, one evaluation of traditional passwords reported authentication rates of 27% after just 3 months (*Zviran & Haga, 1993*). A similar evaluation of Passfaces found authentication rates of 72% after 5 months (*Valentine, 1998*). Such studies contribute to the general finding that memory decay impacts image recognition less than it impacts password recall. Here we show that memory decay impacts face familiarity judgements even less. Second, it does not matter greatly if authentication is observed. As Study 2 shows, even when an attacker sees the same set of targets when attempting to authenticate, authentication is still difficult when different photos of those targets are presented. Previous work has shown that Passfaces is highly vulnerable to

**Table 5 Threat model.** A threat model for Facelock, based on *De Angeli et al. (2005)*.

| Threat | Vulnerability | Attack exploits | Facelock mitigation |
|---|---|---|---|
| Guessability | Predictable choices | Knowledge of a user | Targets are minor celebrities |
| Observability | Ease of shoulder surfing | Observation of user selecting faces | Different images of different targets for each login |
| | Ease of intersection attacks | Refreshing the screen to see which face stays the same | Different images of different targets at each refresh Limited login attempts |
| Recordability | Ease of recording targets' names | User insecure behaviour | No incentive for account holders to write down target names |
| | Ease of recording the screen | Use of mobile phone cameras or screen shots | Different images of different targets for each login |

shoulder surfing when a mouse pointer is used to select targets. Participants in that study rated the vulnerability of Passfaces at 5.2 on a scale from 1 (not vulnerable) to 7 (extremely vulnerable), indicating that shoulder surfers found it very easy to obtain the faces by observation. In the same study, dictionary based passwords were rated 4.85 in terms of vulnerability. Interestingly, using a keyboard instead of a mouse to select targets reduced the vulnerability of Passfaces from 5.2 to 2.3, presumably because keyboard entry forced onlookers to divide their attention between the screen and the keyboard (*Braun, 1998*). For the same reason, keyboard input should strengthen the scheme we propose here.

## Limitations

Our testing exposed a number of important limitations to the system in its experimental form. First, the lock is vulnerable to an attacker who, like the account holder, knows the target faces. This was evident in Study 1, in which attackers who were closest acquaintances of the account holders correctly guessed more targets than attackers who were less close acquaintances. This vulnerability underscores the importance of appropriate target selection. One way for a secret holder to minimise risk would be to maintain a large pool of target faces, and to sample these from disparate fields of interest, so that no single attacker knows enough targets to authenticate.

A second limitation is that attackers may be able to match different images of targets whose appearance is both distinctive (e.g., bald head and round glasses), and stable (i.e., similar appearance in all photos). This was seen in Study 2, where one lock that contained highly distinctive faces could be compromised in a shoulder-surfing attack. For similar reasons, target distinctiveness may be a concern whenever an account holder's targets are all drawn from a single ethnic group or age band. These risks could be reduced by avoiding highly distinctive faces, and by avoiding similar images of any particular target.

*De Angeli et al. (2005)* proposed that graphical authentication mechanisms such as Facelock should be assessed in terms of guessability, observability and recordability when considering how they can be breached. Table 5 shows a threat model based on this taxonomy.

## Future directions

One pragmatic concern is scalability. Our experimental implementation of Facelock involved a multi-step enrollment process, and required considerable human labour to find images of targets and verify these with the account holders. This may not be feasible for a large-scale system. Unless these steps can be significantly streamlined, the approach may be better suited to small-scale or personal deployments such as locking computers and mobile devices than to large-scale deployments such as securing bank accounts.

The studies we report here suggest a number of possible directions for future development. One would be to select non-targets automatically for each grid based on their similarity to the target. For example, if the target for a particular grid is a young Asian female, the non-targets used to complete that grid could also be young Asian females. Increasing the homogeneity of the grids should undermine attacks that rely on distinctiveness to infer targets (Study 1). This functionality would require all images in the system to be tagged with properties such as age, sex, and race. Automatic tagging is currently a major focus of image analysis (*Datta et al., 2008*), and much progress has been made in recent years (see *Bengio, 2009*, for an instructive overview). Indeed, human similarity ratings of faces can already be accurately predicted by automatic systems (*Lacroix, Postma & Murre, 2005*), which could dramatically improve the effectiveness of facelock image arrays.

We noted in Study 1 that 80% of account holders would choose different targets if they could choose again. As authentication failures were so rare among these account holders, it seems reasonable to assume that they would not have chosen different targets to make their own authentication even easier, but rather to make fraudulent access even harder. Presumably faces that are less widely known occurred to these account holders after the study had begun, and the account holders realised that these would make better targets. If so, allowing account holders to update their pool of target faces could improve the security of the system.

A related issue concerns the optimal number and set size of the grids that are used to authenticate. In the present studies we arbitrarily chose a sequence of four $3 \times 3$ grids, which corresponds to a guessing rate of 1 in 6,561. It would be technically trivial to change the guessing rate by changing the grid configuration (e.g., 1 in 1,048,576 for 5 different $4 \times 4$ grids), but implementation details are not our priority here. Our main concern is whether familiarity contrasts in face recognition may be exploited to improve the security of authentication systems. This question is independent of any particular grid configuration. Dedicated usability studies will be required to examine trade offs between security and ease of use. Such studies should also seek to optimise task instructions to make them as easy as possible to follow. In Study 1, five of the eleven successful attacks from personal attackers, and the single successful attack from a random attacker, were all attributable to account holders nominating major celebrities as their targets, despite instructions to the contrary. Clearer instructions, or tighter constraints on the target nomination process, could mitigate this vulnerability.

## Concluding remarks

Although we have outlined a novel approach to graphical authentication using faces, there are clearly very many issues outstanding. In this final section we highlight some of these in the hope that we can be as clear as possible in articulating what is and is not claimed for this proposal.

- We are not presenting Facelock as a packaged product that is ready to deploy. Instead we offer these initial studies as proof of principle. Our focus throughout is on the familiarity of a face to the observer, and how this profoundly affects the observer's ability to process images of that face. The key contrast between familiar and unfamiliar face perception has seldom been addressed in the computer science literature (*Sinha et al., 2006*). Here we hope to have demonstrated that this contrast may be usefully exploited in graphical authentication systems. However, a number of usability issues (discussed above) would need to be resolved before such a system could be practically deployed.

- We do not claim that the proposed system is flawless. In the studies we present, some account holders failed to authenticate, and some attackers succeeded. We address both of these outcomes, alongside other limitations of the studies, in the discussion section of the paper. Our main emphasis is the relative performance of observers who are familiar or unfamiliar with the faces concerned. In perceptual experiments, recognition performance is radically different for these two groups. Here we show that the same applies when the task is incorporated in an authentication system.

- We are not claiming that Facelock is superior to Passfaces. Any such evaluation would require a direct comparison of the two approaches, and we have not attempted that here. Previous studies have looked at memorability of Passfaces (*Valentine, 1998*) and its susceptibility to shoulder-surfing attacks (*Tari, Ozok & Holden, 2006*), and we consider these issues also. However, Passfaces is an established commercial system. Facelock, as an experimental proposal, is unfettered by implementation concerns. Any attempt to compare performance directly would thus be rather unfair on Passfaces. Indeed, the general question of which system authentication system is 'best' is likely too simplistic. Any approach will have its own profile of strengths of weaknesses, and will be better suited to some situations—and to some users—than to others.

- We do claim that it is easy for users to generate a set of faces that are well known to them, but not to other people. We show that an authentication code based on such faces makes it easy for the user to login, even after a year of disuse, as it does not require the user to commit anything to memory. The user's authentication code is difficult for other people to guess, even for close acquaintances such as spouses. It is also highly resistant to shoulder-surfing, as image changes that are transparent for the (familiar) user are not transparent for the (unfamiliar) attacker.

More generally, we propose that research into graphical authentication systems can exploit findings from psychological research, and that psychological research can be enriched by considering applied problems in other fields. Image recognition is not the same as

face recognition. Unfamiliar face recognition is not the same as familiar face recognition. Not all observers are equal. These insights offer much scope for innovation in face-based graphical authentication systems, and we hope that the current studies might spur further development in this direction.

### Funding
This work was supported by an ESRC grant (060-25-0010) to Rob Jenkins and a Wellcome Trust Vacation Scholarship to Jane McLachlan. The funders had no role in study design, data collection and analysis, decision to publish, or preparation of the manuscript.

### Grant Disclosures
The following grant information was disclosed by the authors:
Economic and Social Research Council UK ESRC 060-25-0010.
Wellcome Trust Vacation Scholarship.

### Competing Interests
The authors declare there are no competing interests.

### Author Contributions
- Rob Jenkins and Karen Renaud conceived and designed the experiments, analyzed the data, contributed reagents/materials/analysis tools, wrote the paper, prepared figures and/or tables, reviewed drafts of the paper.
- Jane L. McLachlan performed the experiments, analyzed the data, contributed reagents/materials/analysis tools, reviewed drafts of the paper.

### Human Ethics
The following information was supplied relating to ethical approvals (i.e., approving body and any reference numbers):

FIMS faculty ethics committee, University of Glasgow: Approval number CSE 00871.

### Supplemental Information
Supplemental information for this article can be found online at http://dx.doi.org/10.7717/peerj.444.

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
