# Peer review of "Facelock: familiarity-based graphical authentication"

_PeerJ, doi:10.7717/peerj.444_

## Round 0.1 · original submission · Minor Revisions

I read the manuscript with great attention, and in view of the fact that it deals with the rather sensitive issue of security, I decided to have it reviewed by three experts in face perception and related aspects. My first impressions before their reviews had been returned were that the idea behind the study is brilliant, and that the system that could be implemented based on that idea appears very promising, both as a specific application of cognitive psychology to "secure access engineering", and as a nice example of smart interdisciplinarity. As you will see from their reviews, all of the referees basically share the same impressions. Apart from minor requests, that range from better framing the manuscript in the context of the life sciences to suggesting references on similarity tagging that might be useful for future developments, the general picture is that your manuscript passed the peer review check unanimously, hence I congratulate with you for your work so far and invite you to bring the suggested changes in a revised version.

·

Basic reporting

This is possibly the easiest review I have ever done. I have virtually no suggestions for improvement of this paper!

The paper presents a novel use of known results concerning familiar and unfamiliar face recognition to propose a potential password-less access system. In so doing, they demonstrate the usefulness of these scientific results. It is well-known that people are much better at matching or recognizing familiar faces in multiple images than unfamiliar faces. Hence the idea is for subjects to choose minor celebrities that they know quite well, but are likely to be unfamiliar to others. Then multiple images are found on the internet and used in a 3x3 display, where the user has to simply pick out the familiar face. This is repeated three more times, making the chance of randomly cracking the “Facelock” 0.015%.

They show several novel results, which can be seen as improvements over previous methods:
1. Subjects do not have to remember anything about the system, except to recognize familiar faces. This is demonstrated in their experiment 1, where 97.5% of subjects were able to successfully pass the four challenges (picking the familiar face out of 9), and 84% did so on the first attempt. Furthermore, 86% of subjects were successful a year later, without having interacted with the system at all in between. This is really remarkable!

2. Attackers unfamiliar with the subject were unable to crack the system, except once, when the target subject did not follow instructions and chose famous faces.

3. Personal friends or family were only able to crack the system in 6.6% of the cases.

4. The failure modes, where attackers were able to guess the right faces were interesting: either the subject did not follow instructions and chose famous faces, or subjects chose very distinctive faces, that were very different from the other 8 in the display. This latter problem can easily be fixed by choosing foils that are more similar to the distinctive face.

5. They also demonstrated the robustness of the system to “over the shoulder” attacks. They did this by presenting potential attackers the sequence of choices in the four displays, giving them time to memorize the faces. Then they were shown four new displays, using different pictures of the target faces. Even under these ideal conditions, only 3/160 attacks were successful, and these were all on one out of five tested locks, where the subject had chosen distinctive faces, and the foils were very different.

The only critique I can think of is that in the discussion of future work, where they discuss how to choose similar faces as foils (e.g., if the target is an Asian female, the distractors should be Asian females). They suggest that metadata could solve this problem, but they opine that automatic tagging of faces is not possible using current technology. I don’t believe this is the case. First, new approaches to image recognition, using deep neural networks, are likely to provide this capability (see work by Rob Fergus, Yann LeCun, and Geoff Hinton). Second, there exist systems that can measure similarity between faces that are as correlated with human ratings as humans are with each other (see one of these papers, I can’t remember which one:

Joyca Lacroix, Jaap M. J. Murre, Eric O. Postma, H. Jaap van den Herik: Modeling Recognition Memory Using the Similarity Structure of Natural Input. Cognitive Science 30(1): 121-145 (2006)

Joyca Lacroix, Eric O. Postma, Jaap M. J. Murre: Predicting experimental similarity ratings and recognition rates for individual natural stimuli with the NIM model. BNAIC 2005: 363-364)

Such a system could be used to automatically choose similar faces.

Again, thanks for your very nice paper - it was a joy to read!

Experimental design

I believe all of the work described here could be easily replicated, except the part where one has to wait a year in order to test the success rate of subjects trying to use the system. It was approved by the appropriate human subjects review board.

Validity of the findings

I did not see a way to access the raw data in this paper, but presumably that should not be difficult for the authors to do post-acceptance.

Reviewer 2 ·

Basic reporting

No comments

Experimental design

No comments

Validity of the findings

No comments

Additional comments

I greatly enjoyed reading this manuscript, which explores a novel way of utilizing the different psychological properties of familiar and unfamiliar faces as an authentication code. This distinction is increasingly put to good use in cognitive and forensic psychology to solve some applied problems. However, the current approach – to apply this theorizing to authentication codes – is an interesting idea and was fascinating to read. In a research field that is congested by many variations of the same themes, this is a breath of fresh air. I would certainly like to see this work published and would incorporate it immediately into my own teaching and research programmes.

The manuscript itself is written clearly, the experimental work is rigorous and sound, and the analysis is appropriate. Limitations of the current work are discussed openly and with great honesty. The section on future directions outlines appropriate issues that need to be addressed next. I was also pleased to see the authors address any potential conflicts with Passfaces. It is interesting to note here that Passfaces is based on face learning (familiarization) and therefore fails to tap into the truly robust cognitive presentations of well-established familiar faces. Indeed, it remains unresolved how much learning is required to actually lead to such robust presentations, but we know that this varies across facial identities and observers. For this reason, I would suggest that Facelock is based on a much sounder theoretical footing than Passfaces and the available data suggests as much. It is interesting that the authors have been decidedly moderate about making such inferences in the manuscript – this objectivity and caution is nice to see. Ultimately, this work will draw attention to both approaches, which is great. It will be exciting to see how these face-based authentication approaches develop from here on in.

Reviewer 3 ·

Basic reporting

The paper is well-written and the review of the psychological literature is appropriate. The Authors could give a bit more information about how the Passfaces system is used, its success and in which domains, and highlight to a greater extent than they currently do in what manner the Facelock concept can help to extend and improve the currently existing systems.

Experimental design

No comments.

Validity of the findings

The findings are clear though some of the differences or advantages compared to previous Methods seem overstated. For example, on page 12, the authors state that “returning account holders (68/79) successfully authenticated, 78.5% (62/79) on their first attempt. This is a remarkably well-preserved success rate over such a long period of disuse […] For comparison, previous research reported a lower authentication rate (77% at first attempt) after the much shorter delay of two weeks when using traditional passwords.” However the difference between 78.5% and 77% seems very small to me, even considering the differences in delay between periods of disuse across studies.
It would be useful if the authors could be more explicit about which domains the Facelock (or Passfaces) system can be practical and effective (e.g., authentication coding for mobile phones and personal computers) and which domains are not likely to benefit from this method (Bank/ATMs; credit cards' payments in stores, etc.).

Additional comments

The present study evaluated a novel, still experimental, method of authentication codes based on face identification, as an alternative for passwords and PIN numbers, which are the standard, but are difficult to remember and low in security.

The study is in essence and exploration and extension of an already-existing and market available product called Passfaces: graphical passwords that use faces as a unique verification technology for secure logon. The present Passfaces system uses a two factor authentication process (user-to-site and site-to-user) and it is used in existing security systems (financial and government networks). The adavantage of Passfaces is its intuitive usability and its reliability.

In the present study, the authors reasoned that using a knowledge-based authentication method that does not impose memory load on the user. Given that face identification is a form of perceptual expertise that is general for our species and that the items of expertise can be highly idiosyncratic (who is well-known to me is not necessarily famous for someone else, even siblings and spouses), this can be exploited to relieve processing load. Moreover, as the authors cleverly observe, well-known effects exposed by recent research on face processing can be exploited to make use of faces as authentication codes. A crucial phenomenon is that, despite our ease to recognize familiar faces in highly degraded or very different views than the original or prototypical one, unfamiliar faces are very difficult to match when the viewing conditions are changed. It is this aspect of familiar vs. unfamiliar faces’ recognition that is crucial for developing an authentication code system which is not only resistant to forgetting but also resistant to attempts to break the code (i.e., by providing a personalised ‘facelock’), in the authors’ language, to attack the user’s account. One part of the strategy is then to use a variety of viewing conditions without presenting the same image again and the other essential ingredient is to select faces that have a personalized relation to the account holder instead of being of too much common domain.

The efficacy of this new system was tested in two separate studies, both suggesting that
A method of authentication codes based on face identification is easy terms of memory load and achieves high performance (97.5% success rate), even after a one-year delay (86.1% success). In addition it is robust against attackers or code breakers even from attackers who knew the account holder. The authors therefore conclude that the findings suggest that the contrast between familiar and unfamiliar face recognition is promising for designing new graphical authentication systems.

The manuscript was interesting to read and the idea of a method of authentication codes based on face identification is clever.

I do not have major criticisms, since the studies were well conducted and the authors seem to be well aware of some of the limitations of the proposed system over the traditional passwords and PIN numbers. I list a few points below that may help the author to do some minor revisions to the manuscript.

1) Passfaces is on the market and already exists as a practical graphical method of authentication coding: The authors may want to give a bit more information about how the Passfaces system is used, its success and in which domains, and highlight to a greater extent than they currently do in what manner the Facelock concept can help to extend and improve the currently existing systems.

3) Although I find the study fascinating and the development of the method as an exemplary case of applied psychology, I fail to see the relevance of the present study for the biological sciences (considering that PeerJ is mainly a biology journal). It would be helpful if the authors could justify their studies form such a viewpoint and draw connections with biological (evolutionary) accounts more explicitly than they do.

---

## Round 0.2 · accepted · Accept

Dear Rob and coauthors,

I am happy with your replies to the referees, and with the revised version of your manuscript, so please accept my compliments for your very nice contribution to PeerJ!